# Time Perspective Latent Profile Analysis and Its Meaning for School Burnout, Depression, and Family Acceptance in Adolescents

**DOI:** 10.3390/ijerph20085433

**Published:** 2023-04-07

**Authors:** Joanna Kossewska, Katarzyna Tomaszek, Emilia Macałka

**Affiliations:** 1Institute of Special Education, School Education and Teachers Education, Pedagogical University of Krakow, 30-084 Krakow, Poland; 2Institute of Pedagogy, University of Rzeszów, 35-010 Rzeszów, Poland; 3Rehabilitation and Educational Center, 32-340 Wolbrom, Poland

**Keywords:** student burnout, time perspective, latent profile analysis, depression, COVID-19

## Abstract

This study aimed to apply latent profile analysis (LPA) to (a) empirically identify adolescents’ profiles based on their time perspective (TP), using a person-centered approach; (b) outline the identified profiles across student burnout, depression, and perceived family acceptance; and (c) establish differences between pre-COVID-19 and post-COVID-19 students. Cross-sectional data were collected through an online survey of 668 adolescents. The participants completed the Kutcher Adolescent Depression Scale (KADS), Student School Burnout Scale (SSBS), Time Perspective Inventory (TPI), and Perceived Family Acceptance (PFA) items. Five subtypes of TP were found: hedonistic youth focused mainly on the present time; hedonistic youths focused on the present and future time; fatalistic youths focused on the present and negative past time; future-oriented youths with a positive perception of the past; and hedonistic youths focused on the present with a mild past-negative time perspective. These five profiles were compared for the variables of student burnout, depression, and perceived family acceptance. Scores from SSBS, KADS, and PFA showed a statistical difference across the five subtypes, with the most intensive mental health, social, and educational problems in profile 5. The level of SSBS was significantly different in the pre-COVID-19 and post-COVID-19 samples; however, no significant differences were found in KADS and PFA. Thus, perspective should be emphasized in adolescents with burnout and depression symptoms.

## 1. Introduction

The first case of the novel coronavirus disease (COVID-19) in Poland was reported at the beginning of February 2020 [1], after it was first identified in Wuhan, China in December 2019 [2,3]. The virus spread rapidly and became a major global public health concern. In many countries, as in Poland, governments officially closed schools and universities, implemented closures and limitations in trade and services, and imposed significant movement restrictions on citizens [4].

Enforced remote education was implemented in many countries due to the COVID-19 pandemic, which provided students with an unprecedented and unique experience. Their life situations changed dramatically at all levels, which increased the risk of many basic needs not being satisfied and resulted in numerous new stressors being introduced into their environment [5,6].

Unfavorable conditions in their education, interpersonal conflicts, overload of duties, and the constant stress that accompanies students may lead to a number of negative consequences, such as school phobia, overload, or exhaustion. School burnout syndrome is a long-term reaction to the experience of chronic stress, when students’ personal resources are exhausted and they face simultaneous excessive demands from their environment [7,8,9]. This state generates excessive and intense involvement in professional (or educational) activities, exceeding their energy resources [10]. Burnout syndrome, also referred to in the literature as exhaustion syndrome (analogous to burnout syndrome), consists of three aspects according to the most popular structural approach: a state of chronic energy exhaustion (referred to as emotional burnout or simply burnout), a cynical attitude (referred to as depersonalization), and a sense of dissatisfaction with personal achievements (referred to as a sense of inadequacy) [11,12]. In 2011, Aypay [13] analyzed the symptoms of burnout syndrome among young people and revealed that this phenomenon has a seven-element structure: learning exhaustion, parental pressure burnout, loss of interest in school, homework burnout, teacher pressure burnout, need for rest and free time, and a feeling of ineptitude at school. School burnout is significantly related with interpersonal relationships, which makes students more susceptible to depression and puts them at higher risk of developing suicidal tendencies [12,14].

Globally, the contemporary concept of time perspective (TP) has been widely studied in national and cross-cultural research, because of its wide practical application in various areas of human activity [15]. It is also a core background for linking attitudes, values, mental health, and human behaviors. Zimbardo and Boyd [16,17] defined TP as a non-conscious mechanism by which individuals organize their personal experiences, divided into time zones (past, present, and future), and into five different facets (past-negative, past-positive, present-hedonistic, present-fatalistic, and future). Each perspective is accompanied by a different hierarchy of values and needs, as well as different tendencies of responses to life situations. How one perceives and evaluates the passage of time is an important element of motivation and behavior control [18].

An inappropriate attitude toward time is also associated with increased prevalence of depression. Existing studies show that the greater the intensity of depression, the higher the indicators in terms of the past-negative temporal perspective [19,20]. Considering negative past events and revisiting previous decisions in order to analyze what could have been done differently to improve the situation are important predictors related to the occurrence of depressive disorders [21,22]. This dysfunctional pattern of TPs is measurable and is expressed by deviation from the balanced time perspective (DBTP), which may lead to problems in individual functioning and a higher incidence of depression [22].

Experiencing school stress may lead to school burnout, especially in high-school graduates preparing for their secondary school-leaving (matricular) examinations. Success in these examinations leads to the possibility of continuing higher education. In the COVID-19 lockdown situation, students graduating from high school were unable to prepare properly for their exams, which may have increased their likelihood of developing depressive disorders while resulting in school burnout syndrome.

### The Present Research

Considering the importance of temporal perspective (TP) and student burnout in the mechanism of adolescent depression, we aimed to examine whether students tested before and during COVID-19 differed in their temporal outcome profiles, and whether students who differed in their temporal perspective profiles differed in terms of sense of school burnout, depression, and family acceptance. Particularly, the three main objectives of the current study were to: (a) identify different profiles among adolescents, based on latent profile analysis (LPA) conducted on TP characteristics; (b) outline the identified profiles across school burnout, depression, and perceived family acceptance (PFA); and (c) establish the differences between and among pre-COVID-19 and post-COVID-19 students.

Based on previous works highlighting the developmental sensitivities of TP [23], the distinctive nature of TP in adolescence [24], and the first aim of this study, we hypothesized that at least three to five profiles would need to be found [23,25]. Moreover, we hypothesized that at least two distinguished TP profiles would be characterized by the future-oriented adaptive and fatalistic approaches focused on negative past events and non-adaptive profiles (low future TP/high past-negative TP) (Study hypothesis: H1). Concerning the second study goal, it was expected that students with a future-oriented adaptive profile would be characterized by poor mental health and increased social and educational problems, whereas those with a fatalistic approach focused on the negative past would perceive significantly less family acceptance and experience significantly higher student burnout and depression (Study hypothesis: H2). Regarding the final stated study aim, we expected to find better mental health and fewer social and educational problems among students examined pre-COVID-19 than post-COVID-19 (Study hypothesis: H3).

Considering that TP explains individuals’ ways of framing their thinking, feelings, and behaviors in relation to the past, the present, and the future, the configuration of the TP construct is assumed to be one of the key determinants of higher vulnerability to burnout syndrome [26]. Although studies have rarely addressed the relationship between student burnout and the TP construct, the existing literature suggests that past TP and fatalistic-present TP increase the risk of depression via school burnout [27]. Hence, for a better understanding of the phenomenon and the interplay between the variables of TP construct and student burnout, an additional study aim was included, to test which psychological characteristics more accurately distinguish latent profiles among adolescents. Thus, based on TP and student burnout indicators separately and simultaneously, we examined whether there is an adequate and statistically significant difference in the fit index value in LPA.

## 2. Materials and Methods

### 2.1. Data Collection and Procedure

Participants comprised 668 high school students from several public high schools across Poland, including 415 (62%) girls and 253 (38%) boys. All participants were from the third class in high school, and their ages ranged from 18 to 19 years (M age = 18.50 years). Regarding their COVID-19 pandemic experience, 355 (53%) were categorized as pre-COVID-19 adolescents, tested in April 2019 (Sample 1), and 313 (47%) students were categorized as post-COVID-19 adolescents, who signed up for the survey in April 2020 (Sample 2). All students were White Europeans from different areas of Poland. For sample 1, which comprised individuals examined before the COVID-19 outbreak, paper survey packets were administered among classroom groups of up to 30 students. Participants in sample 2, who were tested during the COVID-19 pandemic, completed the survey online via the Google Forms platform. Participants were assured of anonymity and did not receive any credit for their participation in the survey.

The study was conducted in accordance with the Declaration of Helsinki under relevant local legislation. Permission (No. 07/10/2019) to carry out the research was provided by the Ethical Committees of the Pedagogical University of Krakow for studies involving humans.

#### Instruments

Participants completed several measurement surveys. The Kutcher Adolescents Depression Scale (KADS) is an extensively used screening test to assess the risk of depression in young individuals (Polish adaptation by Mojs) [28]. The KADS consists of six questions or statements regarding everyday experiences, relating to (1) sadness, (2) hopelessness, (3) tiredness, (4) life difficulties, (5) worry, and (6) suicidal symptoms and self-harm. Respondents selected the most suitable answer on a 0–3 scale (0 = hardly ever, 1 = sometimes, 2 = most of the time, and 3 = all the time). A score of 6 points or higher indicates a risk of depression. The reliability of the Polish scale, calculated using Cronbach’s α coefficient, was found to be 0.82. This tool is widely used in North America as a screening test for depression in adolescents [29].

Aypay’s Secondary School Burnout Scale (SSBS) [13] (Polish adaptation by Tomaszek and Muchacka-Cymerman) [30] consists of 34 questions across 7 dimensions. Reliability was calculated using Cronbach’s α: 0.83 for burnout from studying (BFS), 0.82 for burnout from family (BFF), 0.86 for loss of interest in school (LIS), 0.67 for burnout from homework (BFH), 0.75 for burnout from teacher attitudes (BFTA), 0.72 for need to rest and time for fun (NRTF), and 0.72 for feeling of insufficiency at school (FIS). Respondents selected the most suitable answer on a 4-point Likert scale.

The Zimbardo Time Perspective Inventory (ZTPI) [31] (Polish adaptation by Przepiórka) [32] contains 56 statements across five subscales: Past-positive temporal perspective (PP), past-negative temporal perspective (PN), present fatalist temporal perspective (PF), present-hedonistic temporal perspective (PH), and future time perspective (F). Respondents selected the most suitable answer on a 5-point Likert scale. The Cronbach’s α reliability coefficients for the individual subscales in the Polish version were 0.80 (F), 0.72 (PF), 0.72 (PH). 0.61 (PP), and 0.85 (PN).

Respondents also completed the perceived family acceptance (PFA) measure, which consists of a single statement, “I feel safe and accepted in my family”, which was rated on a 5-point Likert scale from 1 (no) to 5 (yes), as well as a demographics survey. Participants were informed that they could withdraw from the study at any time.

### 2.2. Statistical Analysis

LPA was carried out via the R package to identify latent meaningful sub-groups in a dataset, based on continuous variables of TP and SSBS scores. Spurk and colleagues [33] recommend a sample size of around 500 observations for LPA analysis, allowing the researcher to obtain an accurate number of latent profiles. To meet these requirements our sample consisted of 668 observations. We fitted three latent class configurations to determine the optimal number of latent classes: (1) model 1, with only temporal perspective indicators (TP); (2) model 2, with only student burnout indicators (SSBS); and (3) model 3, with all dimensions simultaneously (TP and SSBS). To evaluate the optimal model of LPA, we used: (1) the Bayesian information criterion (BIC), which is a commonly used and reliable index that allows the simplest model to be obtained, with a lower value of BIC suggesting a better fit; (2) the p values of the parametric bootstrapped likelihood ratio test (BLRT), which is based on a bootstrap resampling method, to calculate the significance of the generalized likelihood ratio test—a *p* value equal to or lower than 0.05 indicating a good fit of the model, which means that the corresponding k profile model is better than the k-1 profile model; and (3) integrated information about several fit indices (ICL), which takes into account entropy and allows identification of well-separated clusters (the elbow criterion of ICL refers to the moment when the number of classes increases, with a decrease in ICL, the minimal ICL-favored well-separated clusters) [33,34,35].

To contrast temporal perspective profiles of the selected sub-groups of adolescents (in accordance with the profiles distinguished via LPA), simultaneously taking into account the measurements before and during COVID-19, the results for student burnout, depressive symptoms, and subjective PFA were compared. For this purpose, the two-factor aligned rank transformation (ART) ANOVA was used. (ART) ANOVA is a non-parametric approach to factorial ANOVA, that makes it possible to analyze the main effects and interactions using data ranked before calculation. This statistic was applied because of the non-normal distribution and the inequality of participant numbers in each sub-group. Tukey HSD post-hoc testing was applied to conduct a pairwise comparison analysis. Cohen’s *d* effect size r was calculated as a measure of the distance between distinguished sub-groups.

## 3. Results

### 3.1. Descriptive Results

First, the data were checked to ensure no missing data or anomalies in the cases; then, the mean value, standard deviation, skewness, and kurtosis coefficient were calculated for each variable. There were no missing values detected in the results matrix, and the values for only one score, namely perceived family acceptance (PFA), did not meet the normality criterion; however, the deviations did not exceed the range −2 to +2 (see Appendix A).

### 3.2. LPA Results

The LPA was estimated to determine the number of profiles for students’ temporal perspective (measured with the ZTPI scale, with five indicators): PP, PN, PF, PH, and F; school burnout (measured using the SSBS; seven indicators): BFS, BFF, LIS, BFH, BFTA, NRTF, and FIS; and all these parameters simultaneously: TP and SSBS (12 indicators). Based on the bootstrap sequential LRT for the number of mixture components, five profile models for TP, seven profile models for SSBS, and three profile models for SSBS and TP represented the best solutions, as these results had statistically significant LRT values. The models’ fit indices from the latent profile analyses are presented in Table 1. Model 1 presents the best values for Bayesian information criterion (BIC) and integrated complete-data likelihood (ICL). (The bootstrapped likelihood ratio test results (BLRT) for time perspective and school burnout are given in the Appendix A).

Table 2 and Figure 1 display the results of the profile analysis. The five identified profiles were characterized as follows. Profile 1: hedonistic youths focused mainly on the present time; a total of 334 students (50% of the sample); students in this profile reported moderate past-negative and future orientation. Profile 2: hedonistic youths focused on the present and future time; a total of 169 students (25.3% of the sample); this profile included students who reported no negative thinking about the past or fatalistic thoughts about the present. Profile 3: hedonistic youths focused on the present and negative past events; a total of 114 students (17.1% of the sample); students in this profile reported having strong hedonistic attitudes towards time, low positive time perspective, and mild future orientation. Profile 4: future-oriented youths with a positive perception of the past; a total of 29 students (4.3% of the sample); students in this profile reported having low negative time perspective. Profile 5: hedonistic youths focused on the present with mild past-negative and positive time perspective; a total of 22 students (3.3% of the sample).

### 3.3. Differences between Time Perspective Profiles and Perceived Family Acceptance, Student School Burnout, and Context of Pandemic Experience (Pre-COVID-19 vs. Post-COVID-19)

Table 3 presents descriptive statistics for the data measured with PFA, SSBS and KADS instruments. The (ART) ANOVA statistics presented in Table 4 revealed that the mean levels of all tested psychological characteristics were significantly different among the five sub-groups (*p* < 0.001). Regarding the differences between the profiles in terms of the students’ psychological characteristics, adolescents from profiles 1 and 2 scored significantly higher in PFA and lower in SSBS, IC, and KADS than adolescents from profile 3. Moreover, those from profile 5 demonstrated the highest levels in SSBS, BFF, NRTF, FIS, and KADS compared to all the other groups (for KADS, an insignificant difference was obtained only compared to profile 3). KADS values in profile 4 were significantly lower compared to the scores for adolescents from profile 3. Adolescents from profile 1 scored significantly higher in BFF and BFH than those from profile 2, but lower than those from profile 5. Group 2 scored significantly lower in BFS and BFH than groups 3 and 5. Group 5 scored significantly higher in BFH than group 3. Groups 1 and 2 scored significantly lower in BFTA than groups 3 and 5. A significantly lower value for BFTA was also obtained in group 4 compared to group 5. Collectively, the most intensive mental health, social, and educational problems were found in groups 3 and 5; the lowest were found in group 4. Groups 1 and 2 reported moderate values in PFA, SSBS, and KADS.

The results of the aligned rank transformation ANOVA are shown in Table 4. The comparison analysis between pre-COVID-19 and post-COVID-19 adolescents revealed significant differences only in school burnout indicators (*p* < 0.001 for SSBS, BFH, NRTF, and FIS; *p* = 0.003 for BFS; *p* = 0.023 for BFTA). Students examined after the COVID-19 outbreak were significantly more burned out than the pre-COVID-19 students. The Cohen’s *d* effect sizes were large (>0.80) for all variables, except loss of interest (LIS), where the effect size was medium.

The results of the interaction effects for the main examined characteristics are shown in Table 5. Significant interaction effects i.e., profiles vs. COVID-19 experience, were found in all tested variables. The detailed analysis of the differences in time perspective revealed: Profile 1–4: significantly higher scores in all temporal indexes (e.g., PP. PN. PF, F) and lower in PH among students from post-COVID-19 sample; Profile 5: significantly higher scores in PP, PH, and lower in PN, PF, and F among students from post-COVID-19 sample (see Appendix A).

Students examined post-COVID-19 scored higher in perceived family acceptance in profiles 1–4, but lower in profile 5, than those from the pre-COVID-19 sample (see Appendix A). The pre-COVID-19 students scored lower in student school burnout scale than those from the post-COVID-19 sample in each profile (see Appendix A). Regarding adolescent depression, the pre-COVID-19 sample in profile 1 scored higher in KADS than post-COVID-19. In profile 2 the results were similar for both samples. In profiles 3–5 post-COVID-19 youth scored higher than pre-COVID-19 students (see Appendix A).

## 4. Discussion

This study aimed to (a) examine profiles of students regarding their TP; (b) test whether PFA, student burnout, and depression differ between the distinguished student profiles; and (c) examine differences among students examined before and during the COVID-19 outbreak. 

We hypothesized that three to five TP profiles would be identified among adolescents. The results revealed five student profiles identified according to their TP; however, three of these were characterized by hedonistic attitudes towards time as the dominant facet (profiles 1, 2, 5); profile 3, in addition to showing the highest hedonistic time perspective, was also characterized by a high present-fatalistic time perspective. These profiles are named as follows: profile 1—hedonistic youths focused on the present (the most frequent); profile 2—hedonistic youths focused on the present and future; profile 3—hedonistic youths focused on the present and negative past, profile 4—future-oriented youths with a positive perception of the past; profile 5—hedonistic youths with a mild past-negative and positive time perspective (the most rare). These results align with research highlighting the present hedonistic time perspective with a hedonistic, risk-taking attitude to life as a typical characteristic of the adolescent period as it represents identity exploration [36].

We hypothesized two contrary profiles: future-oriented adaptive, and fatalistic-past-negative non-adaptive. The findings partially confirmed study hypothesis H1: we found future-oriented to be the most adaptive profile, represented by only 4.3% of the tested adolescents (profile 4); two non-adaptive profiles, i.e., fatalistic youth focused on the past, represented by 17.1% of participants (profile 3); and hedonistic adolescents with a moderate past time perspective (profile 5), represented by 3.4% participants. Profile 1 (50% of tested adolescents) indicated moderate adaptability, as these students were less focused on the future, more hedonistic, and more fatalistic than youths from profile 2 (25% of tested adolescents) and the most adaptive profile 4. Additionally, the different TP profiles are distinctively related with students’ outcomes in terms of mental health (i.e., depression) and social (i.e., family acceptance) and educational (i.e., student burnout) problems. Specifically, adolescents from profile 5 were characterized by the highest scores on student burnout and depression, and the lowest scores on PFA. Those from profile 3 experienced a significantly lower level of student burnout but scored similar results in depression and PFA. Adolescents from profiles 1 and 2 were less burned out and depressed than those from profiles 3 and 5. Therefore, study hypothesis H2 was partially confirmed. Furthermore, we also examined the differences in family acceptance, student burnout, and depression between pre-COVID-19 and past-COVID-19 adolescents (study hypothesis H3). Study hypothesis H3 was also partially confirmed by the analysis: post-COVID-19 adolescents scored significantly higher only in student burnout characteristics. Finally, according to these results, the statistical LPA conducted on the TP indicators allows for a better profile model with better fit indices than those performed on student burnout indicators and simultaneously calculated scores for TP and SSBS.

The results obtained in the present study regarding the adaptive vs. non-adaptive TP profiles are consistent with those obtained previously by Dabrowska [37]. The positive role of a future perspective and positive family relationships (equivalent in our study to a sense of acceptance) for reducing stress and distraction symptoms in adolescents is revealed by minimizing school burnout. However, it is worth noting here that, to the best of our knowledge, the LPA used in this article has previously been used in the context of TP only by Braitman and Henson [24], who described TP profiles that protect against risky alcohol use in college students. Individuals with such a protective TP profile do not focus too much on negative past experiences and nor on seeking pleasure, and are focused on the future. Students with such temporal characteristics, compared to students with characteristic profiles that are less protective, consume alcohol far less often and in much smaller amounts. A highly future-focused perspective is thus an important component of the most protective TP profile described by Braitman and Henson [24] and is associated with lower present-fatalistic, present-hedonistic, and past-negative perspectives.

Furthermore, several other studies [38,39,40] have shown that future temporal perspective is a protective resource against depression, feelings of hopelessness, risky driving, and substance abuse. It is also positively related to self-regulation and goal setting [41] and is a positive predictor of academic achievement at school and university levels [23]. Future-oriented individuals are able to accurately formulate their future goals and anticipate the future implications of current decisions and behaviors [42], which is associated with resilience to distractors and the ability to wait for deferred rewards. These traits imply that future-oriented individuals have high self-esteem, intrinsic motivation to learn, and a strong sense of mastery.

Notably, the recently published study by Zong and colleagues [42] also confirmed the significant role of the future and present hedonistic perspective on the association between perceived adverse COVID-19 impact and self-control, well-being, and ill-being. Interestingly, the findings indicated that the positive impact of having a future time perspective depends on the context, e.g., individuals under severe stress experience negative emotions and anxiety despite their high future time perspective. These results indicate that the direct positive effect of future time perspective may be cancelled by existential threats such as COVID-19 disease. Furthermore, negative or confused beliefs about the future may lead people to reverse behavior in the present. Such conditions may further push young people to socially unacceptable behaviors, and mental health problems, i.e., depression [43,44]

Non-adaptive and risky TP profiles were related to past-negative and hedonistic present attitudes, similar to the research of Braitman and Henson [24], who found that high-risk time perspective profiles in their studies was less frequent in individuals with future perspective and higher in present-fatalistic, present hedonism, and past-negative, reflecting that these groups were not very future-focused, were comparatively defeatist, were pleasure-seeking, and viewed their pasts more unfavorably than did their peers. A past-negative perspective is related to psychiatric problems [45,46,47] and people who ruminate over unhappy memories are more susceptible to depressive and anxiety symptoms [48]. Additionally, Chinese adolescents reporting severe depressive and severe anxiety symptoms tended to embrace a negative perspective of their past, compared with their counterparts [49]. The non-adaptability of the negative view of the past and the fatalistic present time perspective was also reflected in educational/vocational commitment, interpersonal commitment, and diffuse-avoidant identity style [36].

Thus, our study fills several gaps in the field: (1) the findings address understudied LPA analysis on TP and student burnout concepts among adolescents; (2) the analysis uses LPA on TP and SSBS variables independently and simultaneously; (3) the discussion examines TP profile differences in mental health and social and educational problems in the context of the COVID-19 pandemic. The current study also adds to the existing literature on student burnout by exploring an important personal resource that contributes to adolescent health and development. Practitioners should develop and implement programs to enhance students’ future time positive orientation, purpose in life, and pursuit of goals; making them less burned out and depressed. Moreover, relevant interventions should be considered, especially among students who are forced to use distance learning and undergo social isolation or are exposed to life-threatening conditions caused by a pandemic, or for other reasons, e.g., temporary isolation or health problems.

### Limitations and Future Directions

The study has some limitations. Identification of the most and least predictive TP profiles for depression and school burnout in students should be based on a larger sample of randomized control studies, to allow a more effective generalization. Hence, it is recommended to replicate the study using more restrictive and controlled procedures for recruiting study participants and data collection. Furthermore, the online data collection procedure also had many limitations [50], and comparing pencil–paper vs. online data could have created an additional burden of inconsistency.

Moreover, statistically derived temporal perspective profiles may not reflect clinically relevant differences in depression and student burnout. To learn more about whether the distinguished profiles are significant, future research should examine long-term mental health and educational outcomes. In addition, future research should explore whether interventions targeting TP can actually shift college students into the more protective profile and decrease depressive as well as burnout symptoms. Finally, TP is not the only one influential individual difference apparently connected with students’ adolescent depression and burnout. Future research should explore the association of TP in the context of other predictors such as personality factors and coping style. Meanwhile, the presented results complement person-centered studies (LPA) related to student burnout before and during the COVID-19 pandemic, and can help researchers and educational practitioners in developing interventions. Institutions can thus devote their limited resources to students who would reap the most benefits. Emerging evidence for a high-risk profile for depression and burnout needs to be further explored.

## 5. Conclusions

The present findings provide indications for potential pathways by which depletion in time-orientation resources may be connected to educational and mental health problems. We identified five profiles of TP: profile 1—hedonistic youths focused mainly on the present time; profile 2—hedonistic youths focused on the present and future time; profile 3—hedonistic youths focused on the present and negative past time; profile 4—future-oriented youths with a positive perception of the past; and profile 5—hedonistic youths focused on the present with a mild past-negative TP. Observed TP profiles were compared for the variables of student burnout, depression, and PFA, and results showed statistical difference across the subtypes, with the most intensive mental health, social, and educational problems for profile 5. Levels of school burnout were significantly higher in post-COVID-19 samples compared with pre-COVID-19; however, no significant differences were found in depression or family acceptance. A protective time-perspective profile might predict mental health as associated with lower depression and lower school burnout. Clarifying TP profiles among students can help identify the students most in need of psychological interventions.

## Figures and Tables

**Figure 1 ijerph-20-05433-f001:**
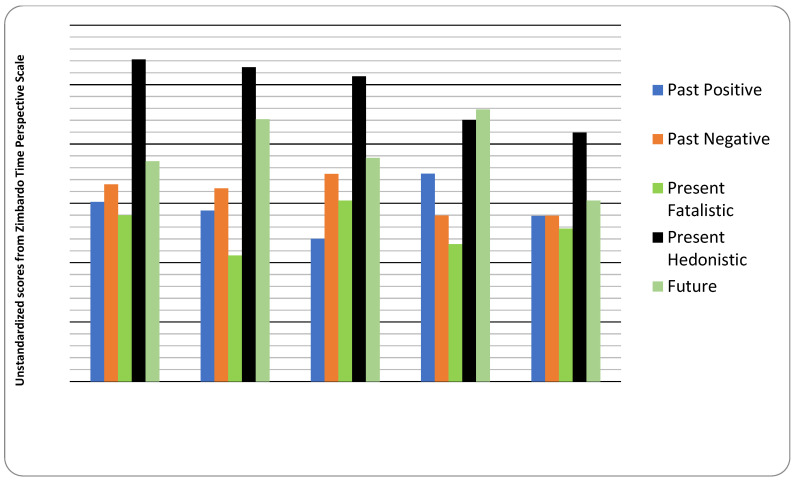
The temporal perspective in five distinctive sub-groups.

**Table 1 ijerph-20-05433-t001:** Fit indices of latent profile models of time perspective (ZTPI scale) and school burnout (SSBS scale) profiles (*N* = 668).

Fit Indexes	Models		
	1. Time Perspective (ZTPI)	2. Student Burnout (SSBS)	3. Time Perspective and Student Burnout (ZTPI & SSBS)
Variance estimation type	VVE	VEI	VEE
*G*	5	7	3
*LL*	−10,655.36	−11,875.89	−22,600.77
*df*	64	68	140
*BIC*	−21,727.00	−24,194.07	−46,112.14
*ICL*	−22,035.83	−24,454.74	−46,271.18

Note: VVE—varying volume and shape, equal orientation; VEI—varying volume, equal shape, and undefined orientation; VEE—varying volume, equal shape and orientation; *G*—number of mixture components (profiles), *LL*—log likelihood of the data; *df*—model’s degrees of freedom; *BIC*—Bayesian information criterion; *ICL*—integrated complete-data likelihood.

**Table 2 ijerph-20-05433-t002:** Means and standard errors for the five profiles identified through LPA (*N* = 668).

Variables	Profile 1 (*N* = 334)		Profile 2 (*N* = 169)		Profile 3 (*N* = 114)		Profile 4 (*N* = 29)		Profile 5 (*N* = 22)		(*ART*) F	Post Hoc
	Mean	SD	Mean	SD	Mean	SD	Mean	SD	Mean	SD		
PP	30.22	4.34	28.78	5.71	24.03	5.65	34.97	0.78	27.86	2.49	34.95 ***	[1–3;1–4;2–3;2–4;3–4;4–5]
PN	33.18	6.32	32.51	8.93	34.95	7.75	27.93	2.05	27.91	4.26	5.36 ***	[1–4;3–4;3–5]
PF	27.98	3.06	21.23	3.45	30.46	5.13	23.14	3.11	25.73	0.63	155.64 ***	[1–2;1–3;1–4;2–3;2–5;3–4;3–5]
PH	54.25	5.56	52.93	6.99	51.37	12.43	44.03	3.31	41.91	2.18	14.40 ***	[1–2;1–3;1–4;1–5;2–4;2–5;3–4;3–5]
F	37.07	5.70	44.14	6.93	37.63	11.36	45.79	2.70	30.45	3.20	47.66 ***	[1–2;1–4;1–5;2–3;2–5;3–4;3–5;4–5]

*** *p* < 0.001. Note: (*ART*) F—Aligned rank transformation ANOVA, post hoc—results of Tukey HSD statistic, the numbers in brackets are the numbers of profiles for which scores are significantly different; positive temporal perspective (PP), past-negative temporal perspective (PN), present fatalist temporal perspective (PF), present-hedonistic temporal perspective (PH), and future time perspective (F).

**Table 3 ijerph-20-05433-t003:** Means and standard errors for the five profiles identified through LPA, for perceived family acceptance, school burnout, and depression (*N* = 668).

Variables	Profile 1 (*N* = 334)		Profile 2 (*N* = 169)		Profile 3 (*N* = 114)		Profile 4 (*N* = 29)		Profile 5 (*N* = 22)		(*ART*) F	Post Hoc
	Mean	SD	Mean	SD	Mean	SD	Mean	SD	Mean	SD		
PFA	4.44	0.04	4.40	0.06	4.18	0.08	4.79	0.08	4.18	0.20	5.44 ***	[1–3;2–3]
LIS	14.85	0.23	15.65	0.38	17.17	0.47	15.52	0.68	16.64	0.83	7.16 ***	[2–3]
BFS	15.98	0.20	14.57	0.31	16.36	0.39	16.38	0.81	22.64	1.04	7.90 ***	[1–2;1–3;1–5;2–3;2–5;3–5]
BFF	13.03	0.21	12.35	0.36	13.59	0.44	11.69	0.40	19.00	0.60	5.64 ***	[1–5;2–5;3–5;4–5]
BFH	14.75	0.19	13.39	0.30	15.54	0.36	16.21	1.05	20.18	0.68	12.07 ***	[1–2;1–5;2–3;2–5;3–5]
BFTA	10.26	0.16	10.12	0.25	11.36	0.32	9.10	0.41	13.91	0.45	6.48 ***	[1–3;1–5;2–3;2–5;3–4;4–5]
NRTF	11.35	0.17	11.12	0.27	12.13	0.34	11.21	0.76	15.82	0.35	5.17 ***	[1–5;2–5;3–5;4–5]
FIS	12.12	0.17	12.12	0.28	13.31	0.35	11.93	0.38	16.77	0.39	8.56 ***	[1–3;1–5;2–3;2–5;3–5;4–5]
SSBS	92.34	0.89	89.33	1.38	99.45	1.81	92.03	3.44	124.95	3.11	14.28 ***	[1–3;1–5;2–3;2–5;3–5;4–5]
KADS	4.78	0.22	4.81	0.35	7.84	0.47	2.55	0.36	8.23	0.63	14.15 ***	[1–3;1–5;2–3;2–5;3–4;4–5]

*** *p* < 0.001. Note: (*ART*) F—Aligned rank transformation ANOVA, post hoc—results of Tukey HSD statistic, the numbers in brackets are the numbers of profiles with significantly different scores; perceived family acceptance (PFA); school burnout (SSBS); loss of interest in school (LIS); burnout from studying (BFS); burnout from family (BFF); burnout from homework (BFH); burnout from teacher attitudes (BFTA); need to rest and time for fun (NRTF); feeling of insufficiency at school (FIS); adolescent depression (KADS).

**Table 4 ijerph-20-05433-t004:** Comparison of psychological characteristics between pre-COVID-19 (*n* = 355) and post-COVID-19 (*n* = 313) samples: (*ART*) F statistic results.

Variables	Pre-COVID-19 Sample *M* ± *SE*	Post-COVID-19 Sample *M* ± *SE*	(*ART*) F	*d Cohen*	*Effect-Size r*
PFA	4.52 ± 0.03	4.25 ± 0.06	0.79	5.69	0.94
LIS	15.38 ± 0.20	15.71 ± 0.30	0.03	1.29	0.54
BFS	15.26 ± 0.22	17.01 ± 0.36	4.52 **	5.87	0.94
BFF	14.99 ± 0.17	16.98 ± 0.27	9.21 **	8.82	0.98
BFH	12.00 ± 0.18	14.33 ± 0.27	16.38 ***	10.15	0.98
BFTA	13.53 ± 0.15	16.20 ± 0.25	5.22 *	12.95	0.99
NRTF	9.84 ± 0.13	11.21 ± 0.21	13.76 ***	7.84	0.97
FIS	10.46 ± 0.14	12.82 ± 0.22	27.95 ***	12.80	0.99
SSBS	11.11 ± 0.13	14.01 ± 0.20	22.43 ***	17.19	0.99
KADS	87.32 ± 0.69	101.26 ± 1.18	0.17	14.42	0.99

*** *p* < 0.001; ** *p* < 0.01; * *p* < 0.05. Note: perceived family acceptance (PFA); school burnout (SSBS); burnout from studying (BFS); burnout from family (BFF); loss of interest in school (LIS); burnout from homework (BFH); burnout from teacher attitudes (BFTA); need to rest and time for fun (NRTF); feeling of insufficiency at school (FIS); adolescent depression (KADS).

**Table 5 ijerph-20-05433-t005:** Aligned rank transformation ANOVA of the interaction effect of profiles and COVID-19 experience for main psychological measurements: Perceived family acceptance (PFA), student school burnout (SSBS), and adolescent depression (KADS) (*N* = 668).

Variables	Perceived Family Acceptance (PFA)		Student School Burnout (SSBS)		Adolescent Depression (KADS)	
	Sum Sq	(*ART*) F	Sum Sq	(*ART*) F	Sum Sq	(*ART*) F
Profiles	21,387,937.98	5.44 ***	21,679,710.43	14.28 ***	22,206,830.75	14.15 ***
COVID-19 experience	21,854,006.93	0.79	20,804,165.83	22.43 ***	24,626,170.88	0.17
Profiles × COVID-19 experience	22,292,943.34	3.58 **	23,924,081.84	4.34 **	23,638,245.56	7.52 ***

*** *p* < 0.001; ** *p* < 0.01. Note: Sum Sq—Sum of squares, (*ART*) F—Aaligned rank transformation ANOVA.

## Data Availability

The data presented in this study are available on request from the corresponding author. The data are not publicly available due to ethical reasons.

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
