# Peer review of "Time Perspective Latent Profile Analysis and Its Meaning for School Burnout, Depression, and Family Acceptance in Adolescents"

_ijerph, 2023, doi:10.3390/ijerph20085433_

Round 1

Reviewer 1 Report

The manuscript provides time perspective latent profile analysis and its meaning for school burnout, depression, and family acceptance in adolescents. This article is important from a theoretical and practical point of view for professionals who work with the consequences of school stress, burnout, depression. However, some points are need to be improved. I believe that once revisions are made to the study, it would constitute an important contribution to the literature. I give some comments below.

1.     Please give accepted in the literature values of the criteria that demonstrate the high model quality (Lines 169-171).

2.     You provide descriptive statistics for Zimbardo Time Perspective Inventory results. It would be better to demonstrate descriptive statistics also for the data measured by other instruments used in this study.

3.     What are BDS, BDR, BDH, BDT, NR, IC in Tables 3, 4 and Results section? What is AKCEPT in Table 4? Does “Loss” mean “LIS” in Tables 3, 4?

4.     Notes after Tables 3, 4 only partially correspond to the abbreviations.

5.     Does “Post-CM” mean “Post-COVID-19 M” (Table 4)?

6.     Сolour figure (Fig. 1) would be better for perception than monochrome one.

7.   The number of references within the last 5 years should be significantly increased.

Author Response

The authors would like to thank the Reviewer no 1 for his/her insightful comments, in reference to which the article has been corrected and improved. We hope that any corrections and clarifications will be accepted.

Joanna Kossewska

Reviewer 2 Report

Comments to the authors:

This is an interesting topic, with an introduction section that does not need major changes. However, the rest of the manuscript is very unclear and needs to be extensively revised.

Materials and methods section

1.      How many public schools participated?

2.      How was the sample calculated?

3.      Was the selection of schools random and representative of schools in Poland?

4.      What was the student response rate?

5.      Given that the anonymity of the participants was preserved, how was it verified that the online surveys had not been answered more than once per student? Was there a way to disable the survey URL once the survey was answered and sent?

6.      What was the procedure for recruiting study participants?

7.      How did you ensure that student participation was voluntary? Were the objective, justification, procedures, possible discomfort, and nature of the study explained to them?

8.      Did they sign an informed consent, even if the survey was online?

9.      Did they receive any feedback on how they did in their evaluation of depression, burnout, and other variables of interest? All students had the right to know their results to ask for psychological support if it was the case.

10.  Statistical analysis section. To clarify all the criteria used to detect the correct number of latent classes, the authors must include an explanation of which values (of the Bayesian Information Criterion, as well as the bootstrapped likelihood ratio test and ICL) were considered. Include references that support the choice and description of the fit indices used.

11.  Statistical analysis section. Authors should include Cohen’s d as a measure of the distance between groups and explain the criteria used to declare good separation between classes. Include references about it.

Results section

12.  A Table showing descriptive results should be provided so the readers can verify distributions as well as each score divided by sex.

13.  Table 1: Due to the lack of information on the cutoff values to consider the goodness-of-fit indices of the three models adequate, this table is unclear. I look forward to a full description of fit indices in the Statistical Analysis section because none of the values in this table means anything to a reader.

14.  Lines 203 to 2015: Authors should start the paragraph by calling Table 2 and Figure 1, so the readers can better understand the results at the same time on the paragraph and table/figure and use only "profiles" or "clusters", but not both, because the speech of that paragraph is confusing when reviewing Table 2 and Figure 1. Please apply only one or the other word to refer to the groups in Table 2 and Figure 1.

15.  Lines 203 to 215: Why was profile 3 called "fatalistic youths focused on the present and negative past events" if, according to Table 2 and Figure 1, the dominant characteristic was "present hedonistic"?

16.  Table 3 should be self-explanatory, and it is not. As can be seen, a nonparametric two-way ANOVA was performed that compared the means of psychological characteristics at two levels: 1) among the five subgroups, and 2) between pre- and post-COVID-19. I believe that this table could be shortened or include different information for the following reasons: 1) showing the F statistic is not of practical value to the reader; 2) the final column called Eta_sq had not been explained in the text, so it is not known why it is included there; 3) since in some scales a p < 0.05 was observed between Pre- and post-COVID-19, the authors should better mention which of both groups had the higher average (example: Post- > pre-COVID-19); and 4) in a two-factor ANOVA model, I would have expected the interactions found to be described if they had occurred.

17.  Honestly, I don't understand Table 4. First, the "Variable" column has abbreviations that have not been previously used in the text and that do not specify, not even at the bottom of the box, what they mean (AKCEPT???, loss?? ??, BDS????, what are you talking about? Text in lines 239 to 253 does not correspond with that table. Table 4 is unreadable.

18.  Do the numbers and letters in parentheses in the "Overall" column have any practical significance? I understand that means and standard errors are shown, but I don't understand the utility of including that data there. In fact, in a posthoc analysis I would have expected pairwise comparisons and some interaction plot between the two levels (level 1: the five groups, and level 2: pre- and post-COVID-19) to be included.

19.  The discussion section, line 267: “besides the highest fatalistic time perspective, was also characterized by a high hedonist time perspective”. This sentence is incorrect, because, as it is shown in Table 2 and Figure 1, the highest characteristic is hedonistic.

20.  Discussion section, lines 264 to 290: The authors make assertions whose accuracy cannot be verified in the results they show. Specifically, from line 274, when they say that the H1 is partially confirmed. Given the poor quality of the presentation of the results, the reader can't confirm what was said in the Discussion.

21.  I have read the rest of the discussion and conclusions, and I cannot evaluate them due to the unclear results. Major changes to this section should be considered, because of the changes that the Results section warrants.

Author Response

The authors would like to thank the anonymous Reviewer no 2 for his/her insightful comments, in reference to which the article has been corrected and improved. We hope that any corrections and clarifications will be accepted.

Please see the attachment,

Joanna Kossewska

Reviewer 3 Report

“Time perspective latent profile analysis and its meaning for 2 school burnout, depression, and family acceptance in adolescents”, is a well-developed manuscript. However, following points can help in more clarification;

1.       Methodology is not clearly written in the abstract.

2.       There is a need to develop a separate section for developing hypotheses.

3.       Hypotheses statements are missing.

4.       The results are acceptable.

5.       The discussion section needs the comparison of results with the earlier studies.

6.       The first two references need alignment.

7.       Overall, good attempt. The manuscript may be considered acceptable upon inclusion of the above-mentioned points.

Moreover, it is to note that I have not checked the plagiarism of this document. 

Author Response

The authors would like to thank the anonymous Reviewer no 3 for his/her insightful comments, in reference to which the article has been corrected and improved. We hope that any corrections and clarifications will be accepted.

Joanna Kossewska

Round 2

Reviewer 2 Report

Comments to the authors (2nd round):

I should note that, unfortunately, the mere answer "It was not controlled" given by the corresponding author is not enough to clear up any methodological and ethical concerns. The lack of clarity in the description of the procedures of:

1.      how the questionnaires were distributed among the students (it does express this in the pre-pandemic period, but not during the pandemic); as well as,

2.      the lack of an informed consent that explained to the students the objectives of the study, justification, description of the procedures;

3.      the lack of information on whether the participation of the students in filling out the questionnaire was voluntary and not obliged;

4.      the lack of information on how to recruit participants;

5.      the lack of information on the student response rate;

6.      and the ignorance of how many public schools the questionnaire was distributed in,

are all limitations whose mere and concise statement at the end of the discussion does not justify the imprecision of information. I can understand that it is often difficult, if not impossible, to randomize the selection of participants, but I still can't find information in the manuscript about how they were recruited, so it is not possible to have an idea (at least approximate) about the population from which the sample was obtained and to which the results are intended to be generalized.

With what the author exposed in the Material and Methods section, and in the responses to my observations, I do not consider it possible, as a scientist, to replicate the procedures. This seriously affects the internal validity and thus the external validity of the study.

7.      Statistical Analyses: The requested explanation about the criteria used to declare a good separation between classes was not carried out or supported by bibliography.

8.      Thanks for attending my comments in Results and Discussion. These sections improved.

Author Response

Joanna Kossewska

Reviewer 3 Report

Kindly accept the manuscript.

Author Response

The authors would like to thank the Reviewer for accepting the reply.

JoannaKossewska